# Decoding the Feeling: Investigating the Vibration Used in Sim Racing Steering Wheel Haptic Feedback

**DOI:** 10.3390/s25237307

**Published:** 2025-12-01

**Authors:** Ciara J. Murphy, Mark J. Campbell, Adam J. Toth

**Affiliations:** 1Esports Science Research Laboratory, Lero, The Research Ireland Centre for Software Research, University of Limerick, V94 T9PX Limerick, Ireland; ciara.murphy@ul.ie (C.J.M.); adam.toth@ul.ie (A.J.T.); 2Department of Physical Education & Sport Science, University of Limerick, V94 T9PX Limerick, Ireland; 3Sport and Human Performance Research Centre, University of Limerick, V94 T9PX Limerick, Ireland; 4Centre for Sport Leadership, Stellenbosch University, Matieland, Stellenbosch 7602, South Africa

**Keywords:** tactile feedback, virtual environments, sim racing, force feedback, vibrotactile feedback

## Abstract

**Highlights:**

**What are the main findings?**
One of the first studies to decode vibration frequencies transmitted through a sim racing wheel.Frequencies of 25–30 Hz uniquely linked to vibrotactile feedback.

**What are the implications of the main findings?**
Findings reveal how distinct haptic channels shape frequency transmission in sim.Steering wheel haptic feedback is predominantly and consistently contained within the 0–5 Hz and 25–30 Hz frequency ranges.

**Abstract:**

Background: Haptic technology has long been integrated into simulated (sim) environments to create a sense of realism and improve performance. In sim racing, force and vibrotactile feedback have been implemented into steering wheels to create a more realistic experience. However, little is understood about how these types of feedback convey information to the sim racer. This study aimed to decode the vibration frequencies transferred through the steering wheel to the user and investigate how these frequencies vary as the strength of each feedback channel is manipulated. Methods: Using a Noraxon Ultium EMG accelerometer, the movements of a Logitech G Pro sim racing wheel were recorded whilst four participants completed five clean laps across nine different conditions. During each condition, a combination of force feedback (0 nm, 6 nm, or 11 nm) and vibrotactile feedback (0%, 50%, or 100%) settings were altered. Accelerometer data were pre-processed and Fast Fourier Transforms were performed to allow examination of signal power at frequencies of up to 200 Hz. Two-way repeated measures ANOVAs were performed to investigate differences in power at relevant frequencies across conditions and laps. Results: Wheel motion was predominantly contained within the 0–5 Hz (force feedback and racer input) and 25–30 Hz ranges. No significant differences were seen in 0–5 Hz power between conditions, but the 25–30 Hz range was observed to exponentially increase as vibrotactile feedback was linearly increased. Finally, 25–30 Hz power at a fixed vibrotactile feedback intensity significantly decreased when the force feedback intensity was increased. Discussion and Conclusions: This study decodes the haptic feedback relayed to the user through a sim racing wheel and highlights atypical changes to signal amplitude across various frequency bands when altering force and vibrotactile feedback intensity.

## 1. Introduction

Haptics refers to the interaction between computers and humans through the sense of touch [1]. It involves using mechanical force and vibration to activate the somatosensory system to detect movement, slip, pressure, temperature, and vibration [2]. Haptic technology is widely utilised to support existing visual and auditory feedback (e.g., a gaming controller vibrating to help locate a target), or to replace visual or audio cues (e.g., setting your phone to vibrate for notifications). One area where haptic feedback technology is particularly integral is in virtual reality and simulated environments. Here, force and tactile feedback are used to create a more realistic feeling of ‘presence’ and can allow the user to interact more naturally within a virtual environment [3,4].

Simulators with haptic feedback are often used as educational and training tools, particularly for difficult or dangerous tasks, as is the case with surgical training [5], flight simulation [6], and driving simulation [7]. The aim of integrating haptic feedback in these simulators is to recreate the somato-sensations that would be present when performing the same task in a real-world scenario. Current evidence indicates that haptic feedback can be used to augment performance in a variety of tasks [8,9,10] and users perceive simulators with haptic feedback to be more realistic than those without it [11]. One prominent area where haptic feedback is becoming more prominent is in video gaming, in which the primary goals are to create a more enjoyable experience and improve gamer performance.

Among the various types of video games, simulated (sim) racing has experienced significant recent growth in viewership, usership, and overall interest [12]. This growth in the popularity of sim racing has aligned well with the growing popularity of real-world motorsport and the fact that racing simulators are often used to facilitate real-world motor sport training when access to physical race locations is not possible [13]. Traditionally, racing simulators have been predominantly used by professional motorsport teams to test racing set ups and to help racing drivers learn a new car or circuit. These professional simulators often consist of large, expensive equipment, such as motion platforms, which aim to simulate the movement and acceleration of the virtual car [14]. While such high-end equipment is not accessible to most users, technological advancements in both the hardware and software of commercial systems have made highly immersive sim racing more accessible for the general population. Due to this increase in accessibility, sim racing has become a separate sport, where many racers train for sim racing specifically as opposed to exclusively using sim racing to train for real life racing. Among the most notable of the technological advancements allowing this change has been the introduction of haptic feedback to key components of a basic sim racing setup—namely, the steering wheel.

The steering wheel is a logical component within which to implement haptic feedback [15,16] due to the density of mechanoreceptors in the glaborous skin of the hand to detect haptic feedback [17], and the similarity to real life, wherein steering wheels provide feedback in real cars [18]. Real-world motorsport steering wheels provide haptic feedback to the driver, including resistance when turning corners due to the lateral acceleration of the car. They can also transmit vibrations from the tyres on the road surface, information that can indicate tyre wear and slip [19]. As a sim racing car does not physically move, these lateral accelerations and vibrations are absent and are, therefore, artificially introduced through the sim racing wheel to simulate the haptic feedback one would receive in a real-world racing scenario. Previous research also indicates that handheld haptic feedback can induce haptic motion—the perception of movement through haptic stimulation [20,21]. As most commercial sim racing setups do not include tilting motion platforms which can simulate car accelerations through activation of the vestibular apparatus [22], and the fact that the visual system is less reliable at perceiving acceleration compared to velocity [23], steering wheel haptic feedback may serve as a valuable tool to recreate motion sensations, even in the most basic sim racing setups.

Previous studies in simulated driving have identified the benefits of steering wheel haptic feedback on driver performance [24,25,26], and experience [27] when compared to steering wheels with no feedback. In most of these wheels, force feedback is a low frequency, high amplitude form of feedback provided to the racer, whereby resistance to turning the wheel increases when lateral acceleration is high (i.e., racing quickly through a corner) and decreases when lateral acceleration is low (i.e., racing down a straight). Recently, commercial sim racing equipment manufacturers have introduced an additional high frequency form of steering wheel feedback (e.g., Logitech implementing their Trueforce software; [28]). This feedback is reported to simulate elements such as the engine of the car and the interaction between the tyres and track as the car is moving.

When examining the usefulness of these forms of haptic feedback for sim racing, it is important to consider how humans detect and perceive such tactile information. There are four cutaneous mechanoreceptors found in the glabrous skin of the palms which are responsible for the detection of various stimuli, such as vibration, slip, and pressure [17,29]. Each mechanoreceptor is sensitive to different vibration frequencies and amplitude ranges [30,31,32], which allows for the discrimination of various types of haptic stimuli. Despite haptic technology incorporating feedback across a range of frequencies and amplitudes in sim racing wheels, the specific frequency content of the wheel vibration provided to the user is unknown. By understanding the frequency content of the vibration provided to the user through the steering wheel, haptic feedback can be more efficiently leveraged within the sim racing context to improve user experience, realism, and performance.

### Research Questions

The purpose of this study is to decode the force and vibrational haptic feedback from a sim racing steering wheel to better understand the tactile information relayed to a sim racer when racing. We first aim to identify which vibration frequencies contribute to force and vibrotactile feedback. Secondly, we aim to quantify the variance in vibration feedback content across multiple laps and racers. Thirdly, we aim to quantify how altering the intensity of either force feedback or vibrotactile feedback impacts the frequency amplitudes associated with the respective feedback channel. Finally, we aim to quantify how altering the intensity of either force feedback or vibrotactile feedback impacts the frequency amplitudes associated with the alternate feedback channel.

## 2. Materials and Methods

### 2.1. Participants

Four high-skilled male sim racing participants (28 ± 3.83 years) were selectively recruited for this study from a mailing list consisting of experienced sim racers. Participants were all right-handed and reported an average of 6.75 ± 5.91 years of sim racing experience, and 7 ± 3.46 h of sim racing activity per week on a regular basis. As the study focuses on the more stable mechanical output of the steering wheel rather than the driver’s behaviours, we assessed from pilot research that four participants would provide sufficient data for analysis. Equally experienced sim racers were used to ensure the laps were clean and consistent to reduce the data exclusion rate. The details of the study were explained to participants and informed consent was provided prior to commencing the study. Ethical approval was authorised by the University of Limerick research ethics board, in accordance with the Declaration of Helsinki.

### 2.2. Materials

A Noraxon Ultium EMG accelerometer (Scottsdale, AZ, USA) was attached to the top of a Logitech G Pro direct drive steering wheel (Lausanne, Switzerland). The wheel was attached to a sim racing rig, which included Logitech G Pro load cell pedals, a 2560 × 1440 MSI MPG321QRF-QD monitor (New Taipei City, Taiwan) with a 120.01 Hz refresh rate, and a Sparco Grid Q racing seat (Volpaino, Italy) attached to a Sim Racing Components Pro simulator rig (Volpaino, Italy) (see Figure 1). The experiment was run on a computer with an Intel Core i9-12900K (Santa Clara, CA, USA) processor running at 3.2 GHz, 64 GB of RAM, and a Windows 11 operating system. The machine was equipped with an NVIDIA GeForce RTX 4070 Ti graphics card (Santa Clara, CA, USA).

Participants raced with a BMW M4 GT3 car on Brands Hatch GP track using iRacing software (version 2023 season 4—2023.09.05.03). The Test Drive option was used to control the environment for the study, and car damage was disabled to prevent a crash from impeding the experiment. OBS Studio was used to record gameplay, MoTec i2 Pro v1.1.5 was used to collect telemetry data at 50 Hz, and myoRESEARCH analysis software (Noraxon, Scottsdale, AZ, USA) was used to record accelerometer data at 500 Hz. Based on accelerometer placement, the *x*-axis represented medio-lateral wheel motion, the *y*-axis represented antero-posterior motion, and the *z*-axis represented up–down motion. A Mu converter file was used to translate the iRacing telemetry file to MoTec software (Victoria, Australia).

### 2.3. Procedure

After providing demographic and gameplay information, each participant completed an out-lap and 5 laps as fast as they could for each of 9 force feedback–vibrotactile conditions over a two-hour testing period. The 9 conditions consisted of a combination of 3 steering wheel force feedback settings (0 Nm, 6 Nm, 11 Nm) and 3 vibrotactile feedback settings (0%, 50%, 100%) (see Table 1 for all conditions). Prior to racing a given condition, the participant left the room while force feedback and vibrotactile feedback settings were adjusted, both on the wheelbase and in iRacing, while all other setting options remained consistent. Each participant was instructed to race as quickly and as consistently as possible—i.e., to produce qualifying-level fast laps repeatedly. When participants went off track during a lap, they had to complete another to ensure that 5 clean laps were recorded. The start of every lap and any mistakes (off tracks or crashes) were manually logged in the accelerometery data. The maximum laps completed by a participant for a single condition was 8 laps. All 9 conditions were completed in a randomised order for each participant to reduce the risk of the practice effect influencing alterations in racing behaviour and creating a difference in output between the first and last conditions. Participants were allowed to take a short break between each condition to mitigate fatigue. When participants completed the experiment, all accelerometery data, gameplay screen data, and telemetry data were uploaded for processing and analysis.

### 2.4. Data Processing

OBS screen recordings and MoTec telemetry data were used to determine off track laps and ensure laps were consistent. Data from out laps and laps in which participants went off track were removed.

A custom-built programme using LabView 2013 was used to process the cleaned accelerometery data for each lap. As accelerometery data were recorded in milligravities (mG), these data were first converted to m/s^2^. The data were then filtered using a 2nd order lowpass butterworth filter with a cutoff frequency of 200 Hz. Following filtering, a Fast Fourier Transform (FFT) with the rectangular windowing method was performed to express accelerometery data in the frequency domain, with a resolution of 0.1 Hz and a frequency band of 0.2–200 Hz used to remove artefacts below 0.2 Hz based on the number of samples and above 200 Hz based on our lowpass filter and the fact that frequencies greater than 200 Hz would not be adequately represented based on the Nyquist frequency of 251 Hz.

To determine which frequencies contained significant acceleration amplitude within the entire spectrum, we extracted those frequencies with power greater than 3 standard deviations (3SD) of the mean power of the entire spectrum. We then extracted the peak power within each 5 Hz bin across the frequency spectrum. These steps were performed on the acceleration data for each axis and lap of each condition and averaged across laps (see Figure 2 for an example of FFT and peak power extraction across frequency bins).

### 2.5. Data Analysis

Statistical analyses were conducted on the processed frequency data using IBM SPSS v26. Table 2, Table 3 and Table 4 highlight the frequency bins, which contained a statistically relevant proportion of the total wheel motion across laps within each condition (expressed as mean ± SD). A Shapiro–Wilk test was used to assess the normality of the data within each frequency bin.

Upon initial examination of the frequency–power plots for each axis of motion, it was observed that 66–99% of total power recorded across all axes of motion were contained along the medio-lateral (*x*) axis (see Table 5 and Figure 3). As a result, subsequent statistical analyses focus on *x*-axis data.

To identify which vibration frequencies contributed to force and vibrotactile feedback, we numerically examined which frequencies contained non-zero power between conditions where force or vibrotactile intensity was zero and non-zero. To quantify the variance in vibration feedback content across multiple laps and racers, we reported the standard deviation of power across laps within a given frequency band for each condition.

To address aims three and four, two-way (lap x condition) repeated measures ANOVAs were performed on the relevant frequency bins and associated conditions. Where sphericity was violated according to Mauchly’s test, a Greenhouse–Geisser correction was applied. Specifically, to quantify how altering the intensity of either force feedback or vibrotactile feedback, separately, can impact the power of the various vibrational frequencies present across the different conditions, specific simple effect comparisons were made between force feedback-only conditions (i.e., F0V0, F6V0, F11V0) and vibrotactile feedback-only conditions (i.e., F0V0, F0V50, F0V100).

Finally, to examine whether altering the intensity of force feedback impacts the power of frequencies associated with vibrotactile feedback, we compared the power of 25–30 Hz and 75–80 Hz bins across two groups of conditions, one of which contained conditions where vibrotactile feedback remained at 50% intensity (F0V50, F6V50 and F11V50), and the other in which vibrotactile feedback remained at 100% intensity (F0V100, F6V100 and F11V100). Sidak corrections were applied where multiple comparisons were performed, means ± SD are reported in the results and significance was set to *p* < 0.05.

## 3. Results

The contributing frequencies to the steering wheel vibration with significant amplitudes identified were within the 0–5 Hz, 25–30 Hz, 50–55 Hz, 75–80 Hz, and 100–105 Hz frequency bins. Significant amplitudes within the 0–5 Hz bin were observed in all nine conditions, but amplitudes in higher frequency bins were only detected when vibrotactile feedback was present. Amplitudes within the 25–30 Hz bin and 75–80 Hz bin were present in laps with vibrotactile feedback set at 50% and 100%, but amplitudes in the 50–55 Hz and 100–105 Hz bins were only present in laps where vibrotactile feedback was set at 100% intensity.

### 3.1. 0–5 Hz Range

When comparing the power of 0–5 Hz motion across laps and conditions, no significant main effects of condition (F(4.45,66.82) = 1.945, *p* = 0.106, η^2^ = 0.095) laps (F(4,15) = 0.514, *p* = 0.727, η^2^ = 0.121) or the interaction between condition and lap (F(17.82,66.82) = 0.898, *p* = 0.582, η^2^ = 0.193) were found (Figure 4).

### 3.2. 25–30 Hz Range

A significant main effect of condition was found for the power of 25–30 Hz motion (F(2,30.43) = 176.635, *p* > 0.001, η^2^ = 0.922). There was no significant main effect of lap (F(4,15) = 0.652, *p* = 0.634, partial η^2^ = 0.148), nor was there a significant interaction between condition and lap (F(8.12,30.43) = 0.648, *p* = 0.734, η^2^ = 0.147). Pairwise comparisons of conditions indicated that there was a significant difference in 25–30 Hz amplitudes among all 6 conditions containing vibrotactile feedback (see Table 6).

Specifically, when comparing conditions where force feedback settings remain constant and the vibrotactile feedback increased (F0V50 to F0V100, F6V50 to F6V100 and F11T50 to F11V100), a doubling of vibrotactile intensity setting significantly increased 25–30 Hz amplitude by approximately 4-fold (see Figure 5). When comparing conditions where vibrotactile feedback settings remained constant and force feedback was increased (F0V50 to F6V50 to F11V50, and F0V100 to F6V100 to F11V100), doubling force feedback intensity significantly decreased the 25–30 Hz amplitude (see Figure 6).

### 3.3. 50–55 Hz and 75–80 Hz Ranges

Statistical analysis could not be performed to investigate the power of 50–55 Hz wheel motion across conditions, as there were not enough laps with relevant data to allow comparison (see Table 2).

No significant main effect of condition was found for the power of 75–80 Hz wheel motion (F(2,24) = 1.994, *p* = 0.158, η^2^ = 0.142), nor was there an interaction between condition and lap (F(8,24) = 1.688, *p* = 0.153, η^2^ = 0.360). However, there was a significant main effect of the lap (F(4,12) = 3.549, *p* = 0.039, η^2^ = 0.542), with pairwise comparisons showing that 75–80 Hz power was greater during lap 2 compared to lap 1 (*p* = 0.017) and lap 4 (*p* = 0.005). No significant differences in amplitude were found between other laps.

## 4. Discussion

The purpose of this study was to decode the force and vibrational haptic feedback from a sim racing steering wheel to better understand the tactile information relayed to a sim racer when racing. We have decomposed the haptic feedback signal emitted from a sim racing steering wheel. Most importantly, steering wheel haptic feedback is predominantly and consistently contained within the 0–5 Hz and 25–30 Hz frequency ranges; increasing the strength of vibrotactile feedback settings exponentially increases vibration magnitudes, and increasing the strength of force feedback settings causes the magnitude of the 25–30 Hz vibration channel to reduce. These findings are consistent across all conditions tested and provide a clear insight into how the overall haptic feedback relayed to the racer is altered when force and vibrotactile feedback settings are manipulated.

### 4.1. Contributing Frequencies

Our results indicate that force feedback largely contributes to wheel motion below 5 Hz, while vibrotactile feedback predominantly contributes to wheel motion in the frequency ranges of 25–30 Hz, 50–55 Hz, 75–80 Hz, and 100–105 Hz. Contributing frequency ranges remained consistent across conditions, with amplitude within given frequency ranges altering when feedback strength settings were manipulated. Vibrotactile feedback magnitude was also largely consistent across different racers and laps within a given condition. The frequency ranges detected in the *x*-axis (see Table 2) were largely the same ranges detected for the *y*- and *z*-axes (see Table 3 and Table 4). However, some frequencies along the *y*- and *z*-axes were not detected in all laps for all relevant conditions. Overall, medio-lateral motion (*x*-axis) contained between 66% and 99% of the total power across all three axes of motion (Table 5). As the wheel is mounted to be able to move left to right to be able to steer the virtual car, it is logical that most of the power is seen in this axis.

### 4.2. Force Feedback

It was hypothesised that motion created by force feedback would not exceed 10 Hz. All nine conditions contained power within the 0–5 Hz frequency range, suggesting that force feedback operates within this range of motion. However, there were no significant differences observed between conditions, and similar power was seen in the 0–5 Hz range in the condition with no haptic feedback (F0V0) despite participants anecdotally reporting that it was harder to turn the wheel when racing during conditions with force feedback. These findings may be understood by accounting for the racers’ movements when driving in the virtual environment. As the *x*-axis records the medio-lateral movement of the steering wheel, the accelerometer also recorded the steering inputs of the racer during their laps, which aligns with the biomechanics literature in highlighting that typical human motion remains below 10 Hz [33,34]. Overall, the motion created from force feedback in the F6 and F11 conditions is arguably negated by the racer, who fights against the motion of the force feedback to ensure the car remains on track, making it difficult to separate the motion created by force feedback from the motion created by the racers’ steering movements. To combat this, subsequent research may consider using electromyography to assess muscular effort needed when racing to differentiate between different force feedback conditions or the use of custom race wheels with force transducers in the grip [35].

Although we were unable to determine the acceleration of the steering wheel created by the force feedback at mid or high intensity settings, the fact that no power was observed in higher frequencies above 5 Hz suggests that force feedback contributions remain below 5 Hz. Motion within this low frequency range likely activate the rapidly adapting Meissner afferents, which predominantly and most accurately detect low frequency vibration [31], sudden motions on the skin [36], and slip [37,38], and the slowly adapting type 2 (SA2) afferents, which mainly detect skin stretch and thus can determine the direction of object motion creating the stretch of the skin [39,40].

### 4.3. Vibrotactile Feedback

Power in the 25–30 Hz frequency range was only detected in the six conditions which included vibrotactile feedback at 50% or 100%, regardless of whether the force feedback was present or not (see Table 2), indicating that this component of haptic feedback is due to the presence of vibrotactile feedback. Frequency processing and subsequent analysis revealed that 83–91% of the overall signal created from the vibrotactile feedback was encapsulated within the 25–30 Hz frequency range within each condition, with the remaining power present across the 50–55 Hz, 75–80 Hz, and 100–105 Hz frequency ranges.

When assessing the change in feedback amplitude when the strength of vibrotactile feedback was increased from 50% to 100%, the power within the 25–30 Hz range increased almost 4-fold, indicating an exponential increase in signal power. This is the first time this relationship has been shown and contradicts the assumption that the relationship may be linear. This change was consistent across all conditions when comparing different vibrotactile feedback settings while maintaining a constant force feedback strength (i.e., F0V50 to F0V100, F6V50 to F6V100, and F11V50 to F11V100). Although no amplitude for 25–30 Hz exists in conditions with only force feedback and is largely linked to vibrotactile feedback strength, the amplitude of the 25–30 Hz component decreases significantly when the force feedback strength is increased, even when vibrotactile feedback remains the same (i.e., F0V50 to F6V50 to F11V50, and F0V100 to F6V100 to F11V100). This suggests an interaction effect between both types of haptic feedback, whereby increasing force feedback strength reduces the intensity of the vibrotactile feedback. While the reason for this observation is currently unclear, one hypothesis is that there is an output threshold which is being surpassed when both types of feedback are operational, and a dampener may be acting to reduce strain on the motor within the wheelbase to ensure that the intensity of the total feedback is within a certain range. Future research may aim to examine more settings (e.g., 3 Nm and 9 Nm force feedback, 25% and 75% vibrotactile feedback) to further examine the rate at which the power changes, determining whether there is a consistent power adjustment between all conditions.

When considering racer perception of the vibrotactile feedback provided by the wheel, the contributing frequency ranges identified align closely with the current understanding of human perception of vibration. Rapidly adapting Meissner afferents and corpuscles are most sensitive to frequency vibrations between 10 Hz and 50 Hz and can detect vibrations at these frequencies even at low amplitudes [31,41]. As they are also considered important in the detection of slip and rapid motions on the skin [36,37], the Meissner corpuscles may allow racers to detect vibrations within the 25–30 Hz range. Pacinian corpuscles are considered the primary mechanoreceptors for high frequency vibration detection, sensitive to frequencies between 50 Hz and 300 Hz, with peak sensitivity at around 200 Hz [30,31]. Pacinian corpuscles code lower frequencies with high amplitudes, but can detect high frequencies at very low amplitudes, which may allow racers detect higher frequencies with little power within the vibrotactile feedback.

In research by Brisben and colleagues [30], it was reported that the vibration detection ability of the mechanoreceptors can be enhanced when grasping objects. As a racer’s entire hand grasps the rim of the steering wheel, it is likely that the detection of steering wheel vibrotactile feedback is enhanced in this state, arguably via Meissner and Pacinian signalling. Even in cases where a racer may not have full contact between their entire hand and the steering wheel, as Meissner corpuscles are so widely distributed around the palm and fingers in the hand [41], and as Pacinian corpuscles have such large receptive fields [42,43], racers should still be able to detect the vibrotactile feedback provided, regardless of the way in which they grasp the wheel. One influencing factor, however, may be the force at which they grip the wheel, as increased grip force can alter vibration sensitivity [30].

It is also interesting to note that although it was possible to detect wheel motion up to 250 Hz, no wheel motion was detected at frequencies above 110 Hz. This may be due to research demonstrating that long-term vibration exposure to frequencies over 100 Hz can have negative effects on human hands, potentially damaging skin, blood vessels, and mechanoreceptors [44].

### 4.4. Limitations and Future Directions

As the racers fought against the motions created by the force feedback when turning the wheel around corners, the motion created by the force feedback was negated, and therefore we were unable to definitively record the output strength and motion of force feedback from the wheel. As outlined above, future research may leverage EMG to assess the muscular effort needed to resist the force feedback, something that may lead to improved haptic intensity tuning for performance. While in this study we analysed clean laps by experienced racers, there are numerous in-depth force and vibrotactile feedback settings, in which various elements of the feedback provided can be enhanced or reduced, including filters, dampeners, tyre slip, and engine noise. Moreover, we did not investigate how feedback regarding different surface types (grass, rumble strips in corners, etc.) cars, and environments (wet vs. dry) impact the provision of force and vibrotactile feedback. Further investigation is warranted to isolate the impact that these various elements have on haptic feedback in sim racing to further understand their contribution to overall racer realism and performance. Finally, while this investigation focused on the forces and vibration fed to the user through the steering wheel across 20 laps for each condition, the inclusion of only four participants may limit the generalisability of the findings. Future work should look to investigate the impact of haptic feedback from the racing wheel on racer perceptions and overall performance.

## 5. Conclusions

This study provides the decomposition of a sim racing steering wheel haptic feedback signal. Our findings align with current knowledge regarding effective frequencies for haptic feedback and safe ranges of vibration frequencies for mechanoreceptors in the human hand. Suggested future directions include investigating human muscular efforts against force feedback settings, isolating specific elements within vibrotactile feedback, and examining human perception of these frequencies.

## Figures and Tables

**Figure 1 sensors-25-07307-f001:**
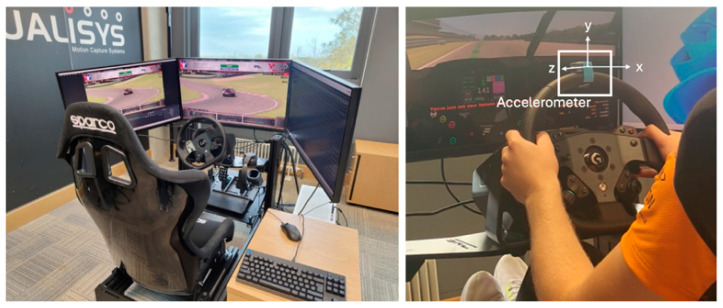
The sim racing experimental setup used for data collection (**left**) and placement of accelerometer (**right**).

**Figure 2 sensors-25-07307-f002:**
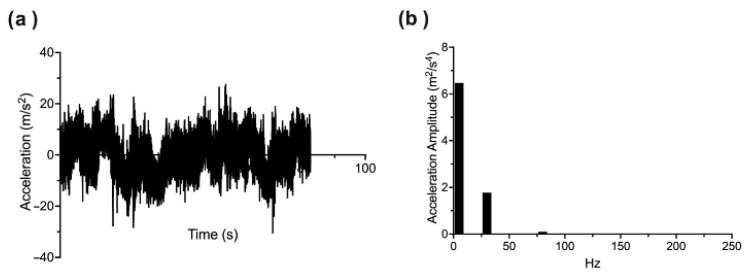
(**a**) time domain visualisation of acceleration recording for one lap. (**b**) Peak amplitudes within 5 Hz frequency windows following a Fast Fourier Transform and the extraction of frequencies with significant amplitude.

**Figure 3 sensors-25-07307-f003:**
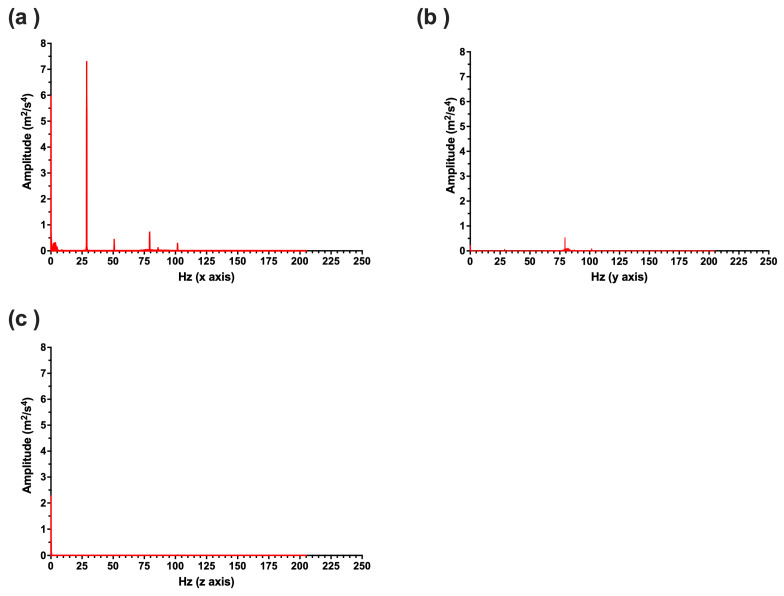
FFT graphs displaying the power spectrum for steering wheel motion along the medio-lateral (x; (**a**)), anterior–posterior (y; (**b**)), and gravitational (z; (**c**)) axes during a lap where force feedback was set to 11 Nm and vibrotactile feedback was set to 100%.

**Figure 4 sensors-25-07307-f004:**
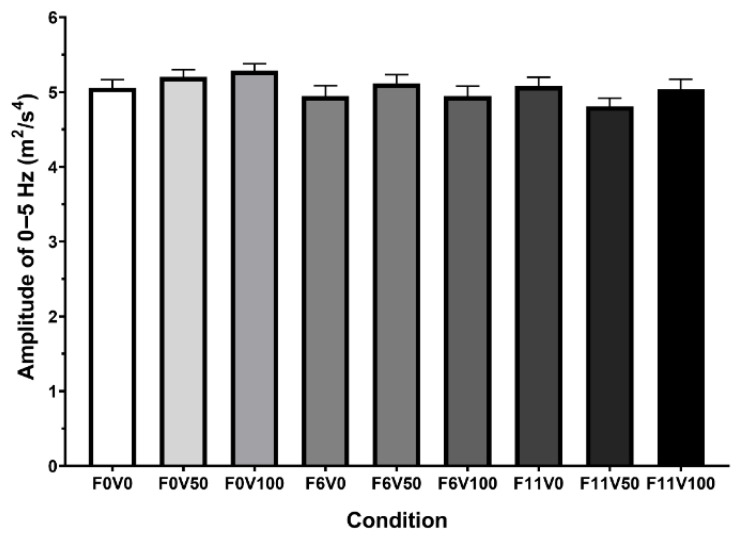
Average amplitude recorded within 0–5 Hz bin for each of the nine conditions. F = Force feedback settings at 0 Nm, 6 Nm or 11 Nm. V = Vibrotactile feedback settings at 0%, 50% or 100%.

**Figure 5 sensors-25-07307-f005:**
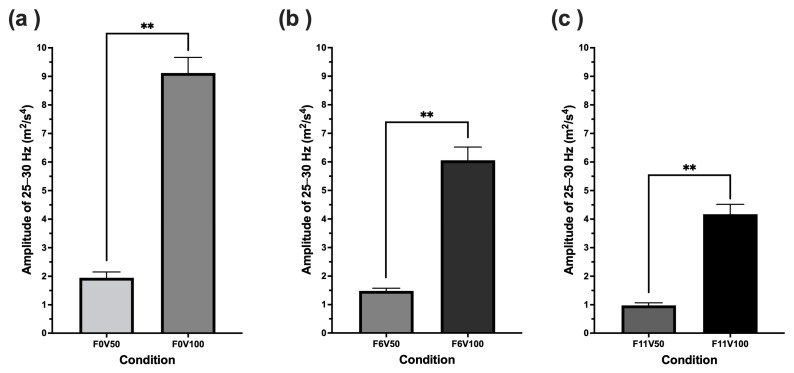
Comparison of 25–30 Hz wheel motion amplitude across V50 and V100 vibrotactile conditions where force feedback was maintained at (**a**) 0 Nm, (**b**) 6 Nm and (**c**) 11 Nm. Darker bars in each graph represent 25–30 Hz amplitudes for conditions including 100% vibrotactile feedback intensities (V100) while lighter shaded bars in each figure represent 25–30 Hz amplitudes for conditions including 50% vibrotactile feedback intensities (V50). ** indicates a significant difference at *p* < 0.001.

**Figure 6 sensors-25-07307-f006:**
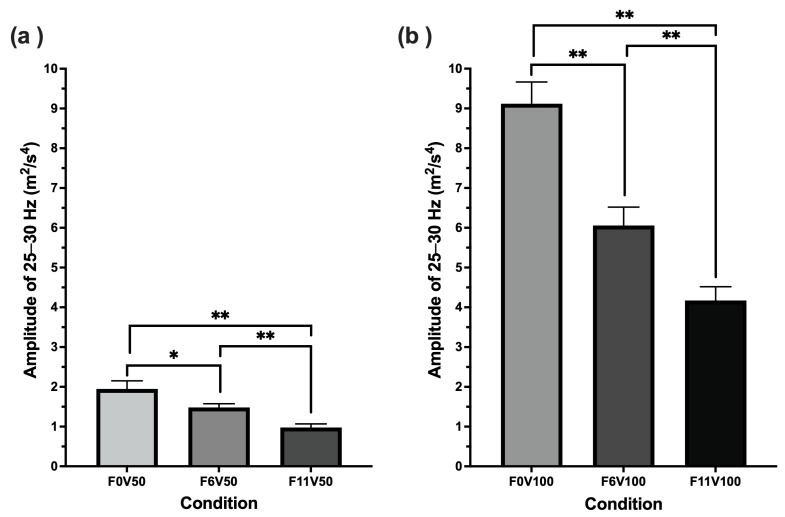
Comparison of 25–30 Hz wheel motion amplitude across F0, F6, and F11 feedback conditions where vibrotactile feedback was maintained at (**a**) 50% and (**b**) 100%. * and ** indicate significance at <0.01 and <0.001 respectively.

**Table 1 sensors-25-07307-t001:** The nine conditions tested showing the combinations of low, mid, and high strengths of force and vibrotactile feedback.

	VFb 0%	VFb 50%	VFb 100%
FFb 0 Nm	F0 V0	F0 V50	F0 V100
FFb 6 Nm	F6 V0	F6 V50	F6 V100
FFb 11 Nm	F11 V0	F11 V50	F11 V100

FFb = force feedback, measured in Nm; VFb = vibrotactile feedback, measured in %.

**Table 2 sensors-25-07307-t002:** Mean ± standard deviation of power (m^2^/s^4^) detected within relevant frequency ranges of medio-lateral (*x*-axis) wheel motion. f0, f6, and f11 refer to force feedback levels set to 0 Nm, 6 Nm, and 11 Nm, respectively. v0, v50, and v100 refer to vibrotactile feedback levels set to 0%, 50%, and 100%, respectively.

Hz	F0 V0	F0 V50	F0 V100	F6 V0	F6 V50	F6 V100	F11 V0	F11 V50	F11 V100
0–5	5.061 ± 0.469	5.205 ± 0.432	5.288 ± 0.400	4.947 ± 0.626	5.116 ± 0.529	4.946 ± 0.596	5.081 ± 0.535	4.808 ± 0.505	5.038 ± 0.604
25–30	-	1.947 ± 0.912	9.118 ± 2.447	-	1.482 ± 0.421	6.058 ± 2.060	-	0.979 ± 0.405	4.172 ± 1.553
50–55	-	-	0.287 ± 0.025 *	-	-	0.304 ± 0.093 *	-	-	0.231 ± 0.094 *
75–80	-	0.182 ± 0.030 *	0.469 ± 0.160 *	-	0.157 ± 0.017 *	0.522 ± 0.193	-	0.152 ± 0.021 *	0.420 ± 0.154
100–105	-	-	0.240 ± 0.036 *	-	-	0.215 ± 0.041 *	-	-	0.201 ± 0.043 *

* indicates where amplitudes were not sufficiently detected for all laps. - indicates where no amplitudes were detected.

**Table 3 sensors-25-07307-t003:** Mean ± standard deviation of power (m^2^/s^4^) detected within relevant frequency ranges of anterior–posterior (*y*-axis) wheel motion. F0, F6, and F11 refer to force feedback levels set to 0 Nm, 6 Nm, and 11 Nm, respectively. V0, V50, and V100 refer to vibrotactile feedback levels set to 0%, 50%, and 100%, respectively.

Hz	F0 V0	F0 V50	F0 V100	F6 V0	F6 V50	F6 V100	F11 V0	F11 V50	F11 V100
0–5	0.157 ± 0.099	0.15 ± 0.712	0.136 ± 0.052	0.133 ± 0.047	0.134 ± 0.053	0.125 ± 0.053	0.132 ± 0.05	0.149 ± 0.068	0.149 ± 0.061
25–30	-	0.03 ± 0.018	0.127 ± 0.06	-	0.02 ± 0.011 *	0.076 ± 0.052	-	0.013 ± 0.003 *	0.034 ± 0.016
50–55	-	0.016 ± 0.004	0.034 ± 0.014 *	-	0.007 ± 0.001	0.027 ± 0.009 *	-	-	0.021 ± 0.007 *
55–60	-	-	0.021 ± 0.005	-	-	-	-	-	-
60–65	-	-	0.021 ± 0.004	-	-	-	-	-	-
65–70	-	0.008 ± 0.003 *	0.021 ± 0.006	-	-	0.018 ± 0.003	-	-	0.015 ± 0.002
70–75	-	0.009 ± 0.001	0.03 ± 0.009	-	0.01 ± 0.002	0.034 ± 0.01	-	0.006 ± 0.001	0.02 ± 0.005
75–80	-	0.052 ± 0.032 *	0.247 ± 0.116 *	-	0.061 ± 0.05 *	0.327 ± 0.151	-	0.039 ± 0.019 *	0.219 ± 0.12
80–85	-	0.02 ± 0.005 *	0.077 ± 0.017	-	0.023 ± 0.009 *	0.077 ± 0.037	-	0.018 ± 0.006 *	0.058 ± 0.027
85–90	-	0.018 ± 0.014	0.045 ± 0.028	-	0.011 ± 0.004	0.031 ± 0.012	-	0.006 ± 0.002	0.023 ± 0.006
90–95	-	0.012 ± 0.003 *	0.031 ± 0.008	-	0.009 ± 0.002 *	0.027 ± 0.006	-	0.009 ± 0.002 *	0.02 ± 0.006
95–100	-	0.01 ± 0.003 *	0.024 ± 0.006	-	0.007 ± 0.00 *	0.018 ± 0.005	-	-	0.013 ± 0.002
100–105	-	0.052 ± 0.051	0.109 ± 0.064 *	-	0.017 ± 0.009	0.062 ± 0.029 *	-	0.011 ±0.004	0.047 ± 0.023 *
105–110	-	-	0.017 ± 0.047	-	-	-	-		0.012 ± 0.001

* indicates where amplitudes were not sufficiently detected for all laps. - indicates where no amplitudes were detected.

**Table 4 sensors-25-07307-t004:** Mean ± standard deviation of power (m^2^/s^4^) detected within relevant frequency ranges of gravitational (*z*-axis) wheel motion. F0, F6, and F11 refer to force feedback levels set to 0 Nm, 6 Nm, and 11 Nm, respectively. V0, V50, and V100 refer to vibrotactile feedback levels set to 0%, 50%, and 100%, respectively.

Hz	F0 V0	F0 V50	F0 V100	F6 V0	F6 V50	F6 V100	F11 V0	F11 V50	F11 V100
0–5	1.352 ± 0.52	1.297 ± 0.34	1.193 ± 0.276	1.182 ± 0.296	1.174 ± 0.305	1.107 ± 0.436	1.165 ± 0.416	1.317 ± 0.606	1.338 ± 0.594
25–30	-	-	-	-	-	0.0285 ± 0.008 *	-	-	-
75–80	-	-	-	-	-	0.0257 ± 0.009 *	-	-	-
100–105	-	-	0.071 ± 0.024 *	-	-	0.03 ± 0.019 *	-	-	0.201 ± 0.043 *

* indicates where amplitudes were not sufficiently detected for all laps. - indicates where no amplitudes were detected.

**Table 5 sensors-25-07307-t005:** Mean amplitude and percent of total power contained within each relevant frequency bin across all laps and conditions along the *x*, *y*, and *z* axes.

Hz	X Mean (%)	Y Mean (%)	Z Mean (%)
0–5 Hz	5.055 (78.59)	0.141 (2.19)	1.231 (19.22)
25–30 Hz	3.959 (97.96)	0.0542 (1.34)	0.029 (0.72)
50–55 Hz	0.266 (91.54)	0.025 (8.46)	0 (0)
75–80 Hz	0.42 (69.67)	0.157 (26.08)	0.026 (4.26)
100–105 Hz	0.212 (66.5)	0.051 (15.48)	0.059 (18.03)

**Table 6 sensors-25-07307-t006:** *p*-Values for 25–30 Hz amplitude comparisons across conditions.

	F0 V50	F0 V100	F6 V50	F6 V100	F11 V50	F11 V100
F0 V50	-	<0.001 **	0.007 *	<0.001 **	<0.001 **	<0.001 **
F0 V100	-	-	<0.001 **	<0.001 **	<0.001 **	<0.001 **
F6 V50	-	-	-	<0.001 **	<0.001 **	<0.001 **
F6 V100	-	-	-	-	<0.001 **	<0.001 **
F11 V50	-	-	-	-	-	<0.001 **
F11 V100	-	-	-	-	-	-

* = indicates *p* < 0.01. ** = indicates *p* < 0.001.

## Data Availability

Data available on request from the authors.

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
