# Peer review of "Decoding the Feeling: Investigating the Vibration Used in Sim Racing Steering Wheel Haptic Feedback"

_sensors, 2025, doi:10.3390/s25237307_

Round 1
Reviewer 1 Report
Comments and Suggestions for Authors
The study aims to decode the vibration frequencies transmitted through a sim racing steering wheel and analyze how these frequencies vary with changes to force-feedback steering wheel force and vibrotactile feedback intensity.
The study has novelty, and the experimental design is clear with well-defined conditions. The use of a high-resolution accelerometer is commended. The paper is well written with clear language and clear figures. Adequate use of references where needed.
The sample size is very small. 4 participants is very low for statistical analysis and especially low for repeated measures ANOVA. This limits the generalizability and statistical power of the paper considerably. This is further damaged by the homogeneity of the participants. Excluding variability is skill level, gender, and handedness can all affect haptic perception. Even car setup, track conditions, and driving styles could have been used to introduce variability.
The study also completely lacks subjective measures, focusing solely on mechanical output and frequency analysis, while user perception might be a very important factor for sim racers.
It was also unclear for me as to why the author chose those vibrotactile feedback frequencies. Pacinian corpuscles respond to a broader range; why not test higher frequencies for which those mechanoreceptors are more sensitive to?
The authors make a case for the steering wheel being the optimal place for vibrotactile feedback for racers. It is my understanding that most racers wear thick gloves while racing.
The authors could try to be more succinct in the presentation. Especially the introduction felt a bit long, perhaps including some unnecessary details.
The term "decoding" in the title seems to suggest such components that were not present in the study. It might be just my understanding of the term, but at least to me, it seemed a bit odd choice. I might well be wrong.
Author Response
Please see the attached responses to reviewers document

Reviewer 2 Report
Comments and Suggestions for Authors
This study analyzes the force and vibrational haptic feedback from a racing steering wheel. Four volunteers participated in the experiments. Authors processed the data and conducted statistical analyses on the results obtained under different conditions.
Abstract
The abstract provides the aim of the study, and explains briefly the methods, key results, and main statistical results. The only comment would be:
Comment 1. Could you please provide the theoretical basis to state that “This study is the first…” (lines: 39-42).
Introduction
The introduction explains key concepts and studies related to the problem.
Comment 2. Authors explain the purpose of the study in the last paragraph of the Introduction (lines 122-130). It would be recommended to add a subsection called “1.1. Contributions of the study”, in which the contributions could be highlighted.
Comment 3. It would be convenient to explain the research questions that the study aims to answer in a subsection “1.2. Research questions”.
Comment 4. It would be recommended to explain the research variables of the study (i.e., the variables that are measured in the study; and the variables that are manipulated in the study) in a subsection titled “1.3. Research variables”
- Materials and Methods
Comment 5. Please enumerate the following subsections: “Participants”, “Materials”, “Procedure”, and “Processing”.
Participants.
Comment 6. It would be convenient to explain the inclusion and exclusion criteria to select the participants. Additionally, it would be recommended to explain whether there was a control group that participated in the experiments. Additionally, it is important to comment on why the sample of participants is too small and what are the implications of this.
Materials
Comment 7. It would be convenient to indicate the software used to process the accelerometer data in this section. Currently, this is explained in line 191.
Comment 8. It would be helpful to add labels to the key elements used in the experiment in Figure 1.
Procedures
Comment 9. Currently, it is mentioned that “All 9 conditions were completed in a randomised order for each participant.” (lines 178-179) Could you please explain why the nine conditions were tested in that specific order, and what implications this might have?
Processing
Comment 10. Please specify that 3SD is three standard deviations to clarify how the frequencies were extracted (line 198)
Data Analysis
Comment 11. Please improve the resolution of Figure 3, so that it can be easier to interpret.
Comment 12. Please provide the alpha used in the statistical tests.
Comment 13. It would be convenient to explain the software used to conduct the statistical analyses in the subsection “Materials”. Currently, it is in lines 219-220
- Results
Authors conducted statistical analyses and explained the results according to the output of the statistical tests. The tables are explained in detail.
Comment 14. Could you please clarify whether: i) any questionnaire was given to participants to gather their opinions on the experience with the sim racing steering wheel ; and ii) any report was obtained regarding the breaks the participants took during the experiments?
- Discussion
Authors explain key findings of their study and describe the main limitation.
Comment 15. The research questions presented in the Introduction should be answered in the discussion section.
Comment 16. It would be recommended to compare the results of the authors’ proposal with results of previous studies.
Comment 17. The sample of the participants is too small. This might be included as a limitation of the study.
- Conclusions
Comment 18. Please provide numerical results to support the key findings (e.g., statistical results).
Acknowledgments, Conflict of Interest, Abbreviations
Comment 19. Please revise the sections: Acknowledgments, conflict of Interest, Abbreviations. Please update them according to the information of your manuscript.
Appendix A, Appendix B.
Comment 20. Please remove these sections since they were not used.
References
Comment 21. Please revise the format of reference 1
Author Response

(The authors gave the same response as above.)

Reviewer 3 Report
Comments and Suggestions for Authors
-
Sample size of 4 participants is far too low. No power calculation or justification. Statistical conclusions are unreliable.
-
Research design weak: single wheel model (Logitech G Pro) only, no test across different hardware, grip force, or mounting conditions. Cannot generalize to sim racing or vehicular sensing.
-
Methods lack rigor:
-
Filtering parameters not fully stated (filter type/order/cut-off).
-
Frequency band selection appears post-hoc; no pre-registered rationale.
-
ANOVA with so few subjects is statistically underpowered and questionable.
-
-
FFT analysis is shallow: no spectral leakage handling, windowing details, or averaging method.
-
Vibration/force manipulation limited to 0,6,11 Nm and 0–100 %. These do not reflect full consumer range or racing forces.
-
Results: 25–30 Hz finding interesting but based on weak stats. No confidence intervals or effect sizes.
-
Discussion overstates novelty; similar signal decoding and haptics papers exist.
-
External validity absent: no mention of road/track surface simulation, temperature, glove use, or driver variability.
-
Data and code not shared; reproducibility is very low.
-
Figures: FFT plots acceptable but lack error bars and subject variability; add per-subject traces.
-
Claims of “first to decode” are exaggerated; at least relate to prior vibration and haptics decoding work.
Author Response

(The authors gave the same response as above.)

Round 2
Reviewer 2 Report
Comments and Suggestions for Authors
Some comments are still unaddressed.
The authors did not explain the implications of a small size of participants and did not report it as a limitation of the study. It is highly recommended to either perform more experiments or at least present this as a limitation and explain its implications.
Moreover, it is important to explain clearly and introduce the research variables of the study as it was suggested in the comment of the previous review report, so that readers could identify them from the very beginning:
Comment 4. It would be recommended to explain the research variables of the study (i.e., the variables that are measured in the study; and the variables that are manipulated in the study) in a subsection titled “1.3. Research variables”
Could you please provide the theoretical basis to state that “For the first time, we have decomposed the haptic feedback signal emitted from a sim racing steering wheel.” (lines: 301-302)
It would be convenient to mention as a limitation or as a future direction that the opinion of the participants will be collected.
Additionally, it is important to support the main conclusions with numerical results as it was recommended in the comment of the previous review report:
Comment 18. Please provide numerical results to support the key findings (e.g., statistical results).
Author Response
We have addressed your concerns in the attached response file

Reviewer 3 Report
Comments and Suggestions for Authors
The revised paper reads much better and feels more polished. Few more obeservations:
- The abstract is informative but slightly long. It can be shortened by reducing repetition and highlighting the main results clearly. Emphasizing the two main frequency bands, 0–5 Hz for force and 25–30 Hz for vibrotactile feedback, will make it more effective.
-
The methods section is detailed and well-structured. The nine-condition design and randomization show good control. It would be better to mention what FFT windowing method was used and whether each lap was analyzed separately or averaged. Also, please clarify if any correction for multiple comparisons was applied.
-
Focusing on the x-axis data is reasonable since it holds most of the total power. It would help to explain briefly why this axis dominates. Adding visible axis labels to Figure 1 will also make it clearer for readers.
-
The results are presented well. Figures 5 and 6 are clear and easy to understand. It would be good to include a note in the captions explaining what the asterisks mean for significance.
-
The discussion connects the findings nicely to human tactile physiology. The explanation linking 0–5 Hz to Meissner receptors and higher frequencies to Pacinian receptors is logical and convincing. This shows strong understanding of sensory science and haptic feedback.
-
The idea that force feedback may reduce vibrotactile output because of a possible threshold is interesting but still speculative. It should be stated as a possible explanation and not a confirmed effect. Future studies can test it using more feedback levels.
-
The limitations are described clearly. Mentioning EMG and other possible tools for future testing strengthens the paper. You could add one short sentence on how this work may help improve haptic tuning for gaming or racing training.
-
The language is clear and professional. A few small typos like “sudy” should be corrected. Notations such as FFb or FFB should be consistent throughout. References should follow the Sensors formatting style.
Author Response

(The authors gave the same response as above.)
